# Neutron Sources at the Frank Laboratory of Neutron Physics of the Joint Institute for Nuclear Research

**Valery N. Shvetsov**

Frank Laboratory of Neutron Physics, Joint Institute for Nuclear Research, 141980 Dubna, Russia; shv@nf.jinr.ru; Tel.: +7-(49621)-6-59-25

**Abstract:** The IBR-2 reactor and IREN facility are the two main neutron sources at the Frank Laboratory of Neutron Physics. This contribution presents data on the IBR-2 reactor parameters before and after modernization. The general schemes of the IBR-2 reactor as well as of the IREN facility are presented.

**Keywords:** high-flux pulsed reactor; IBR-2; IREN facility; LUE-200 electron source

## 1. Introduction

The Joint Institute for Nuclear Research (JINR), which was established in 1956, is a partnership of 18 countries committed to the goal of collective performance of theoretical studies, building and operating of the world's leading facilities for research in condensed matter physics, nuclear physics and elementary particle physics. The two main JINR neutron sources—the fast pulsed reactor IBR-2 and the intense resonance neutron source (IREN—are operated by the Frank Laboratory of Neutron Physics (FLNP), one of the seven JINR laboratories.

## 2. IBR-2 Reactor

The IBR-2—a powerful high-flux pulsed reactor of periodic operation—was put in operation in 1984 at a mean power of 2 MW [1,2]. In 2007, the reactor reached the service life limit on fuel burn-up and fluence on the reactor vessel and was shut down for modernization and replacement of the primary reactor equipment. The main objectives of the modernization were to increase the safety, reliability and experimental possibilities of the reactor for the next 25 years of operation [3]. By 2010 the installation of new equipment was completed and was followed by a successful power startup. The IBR-2 parameters before and after modernization are presented in Table 1.

The main parts of the reactor are shown schematically in Figure 1. The reactor core is an irregular hexagon composed of fuel element subassemblies. Plutonium dioxide pellets are used as a reactor fuel. The cooling system has three circuits and two loops. The IBR-2 is a sodium-cooled fast neutron reactor. The core is installed in a double-walled steel vessel and surrounded by a number of stationary reflectors, control and safety units among them, as well as water moderators serving 14 horizontal beam lines. Water moderators of the reactor thermalize fast neutrons down to a thermal energy range used by experimenters on the extracted neutron beams.

A unique feature of the reactor is the periodic modulation of reactivity, which is accomplished by the rotation of the main moveable reflector and the auxiliary moveable reflector near the core (Figure 2). The rotors of the main and auxiliary movable reflectors rotate in opposite directions with different velocities. At a frequency of 5 Hz the reactor is brought from a deep subcritical state to a prompt supercritical one. A power pulse is generated at the moment when both reflectors approach the core [1,2,4].

**Table 1.** IBR-2 parameters before and after modernization.

| Parameter | Before Modernization | After Modernization |
|---|---|---|
| Average power(MW) | 2 | 2 |
| Fuel | $PuO_2$ | $PuO_2$ |
| Number of fuel assemblies | 78 | 69 |
| Maximum burnup (%) | 6.5 | 9 |
| Pulse repetition rate, Hz | 5, 25 | 5, 10 |
| Pulse half-width, µs: | | |
|    fast neutrons | 215 | 245 |
|    thermal neutrons | 320 | 340 |
| Rotation rate (rev/min): | | |
|    main reflector | 1500 | 600 |
|    auxiliary reflector | 300 | 300 |
| Coolant | Sodium | Sodium |
| Thermal neutron flux density from moderator surface (n/cm$^2$·s): | | |
|    time average | $\sim10^{13}$ | $\sim10^{13}$ |
|    burst maximum | $\sim10^{16}$ | $\sim10^{16}$ |

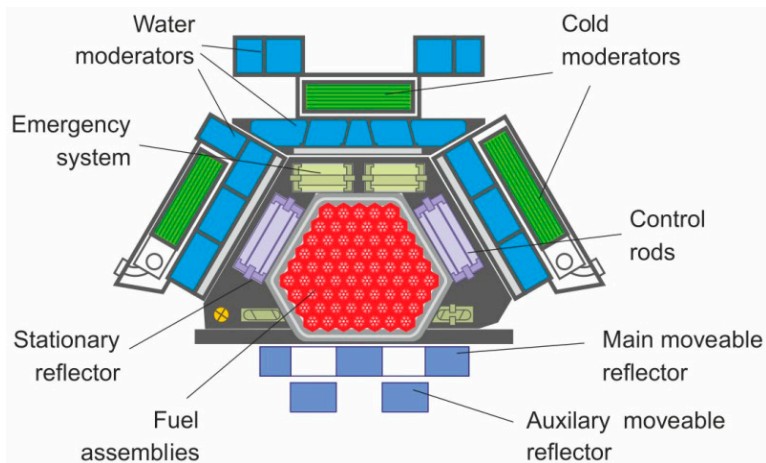

**Figure 1.** Main part of the IBR-2 reactor.

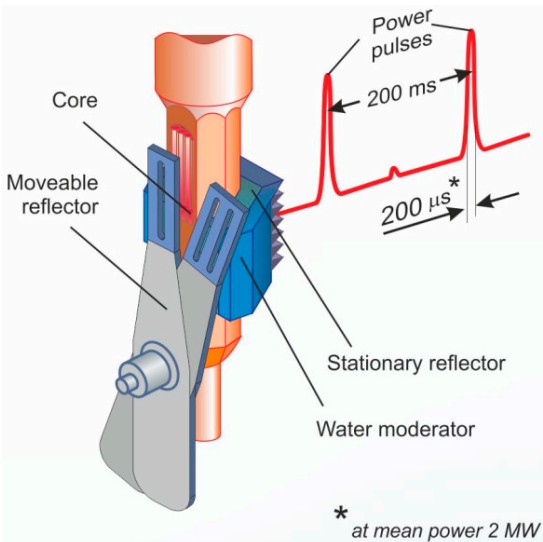

**Figure 2.** Core of the IBR-2 reactor with a movable reflector.

After modernization of the IBR-2 reactor a pelletized moderator based on aromatic hydrocarbons was installed. The main advantages of the moderator based on aromatic hydrocarbons include: low radiolytic hydrogen release, no danger of explosion, obtaining of a cold neutron yield, comparable to that of liquid hydrogen, cheap to develop and safe to operate. Use of this type of moderator allows a substantial reduction in the moderator temperature. A small fraction of the pellets can be periodically discharged and replaced, permitting continuous, arbitrarily long operation of the moderator system [5]. IBR-2 cycles (approximately 2500 h of operation per year) are usually carried out either in the water or cryogenic mode.

The period of reactor modernization was simultaneously used also for upgrading the existing facilities and creating the new ones, which made it possible to increase the number of operating spectrometers from 11 to 14 and to significantly extend the experimental capacities and areas of research carried out (Table 2) [6]. At present, a suite of 13 high-performance instruments (diffractometers, small-angle scattering instruments, reflectometers, inelastic scattering spectrometers) are available for experiments in the framework of the FLNP user program (Figure 3). More than 200 experiments are conducted at the reactor annually by scientists from more than 15 countries. Two other new instruments (a new stress diffractometer and a facility for neutron imaging) are presently under construction. A high fluence irradiation facility for radiation hardness studies and a neutron activation analysis setup are also available for research.

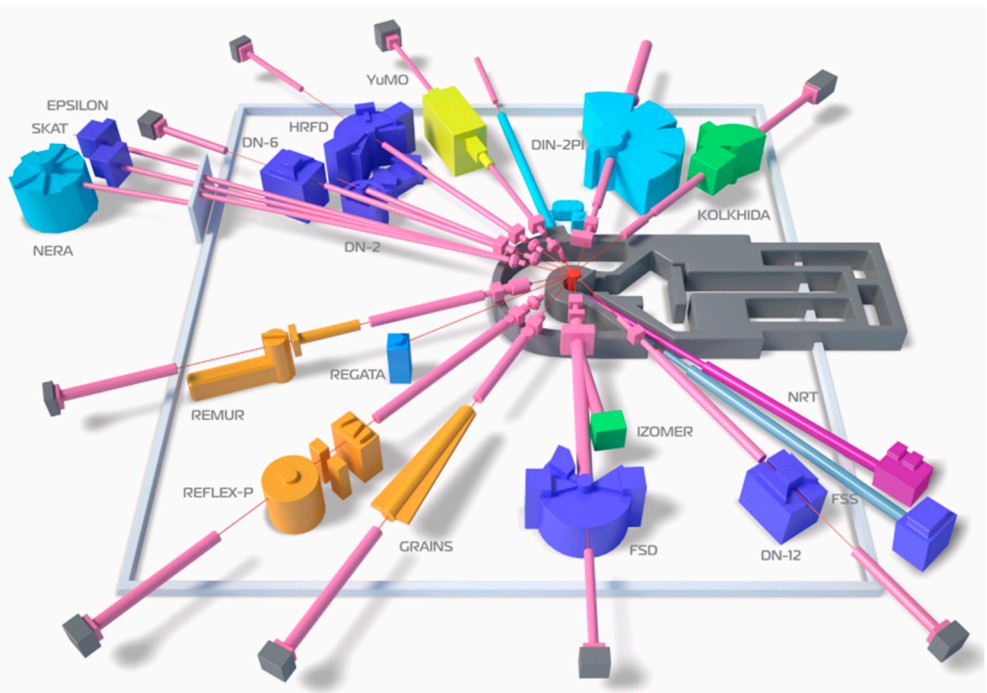

**Figure 3.** Layout of the IBR-2 spectrometer complex.

**Table 2.** List of the IBR-2 instruments.

| № | Name | Type | Domain of research | Reference |
|---|------|------|--------------------|-----------|
| 1. | HRFD | High Resolution Fourier Diffractometer | Determination of structural parameters of crystalline materials with high precision | [7] |
| 2. | RTD (DN-2) | Real-Time Neutron Diffractometer | Determination of structural parameters of crystalline materials and nanosystems (lipid membranes, etc.), real-time studies of chemical and physical processes | [8] |

**Table 2.** *Cont.*

| № | Name | Type | Domain of research | Reference |
|---|------|------|--------------------|-----------|
| 3. | DN-6 | Neutron diffractometer for investigations of micro-samples at high pressure | Determination of parameters of crystal and magnetic structure of materials as function of external pressures | [9] |
| 4. | EPSILON-MDS | The strain/stress diffractometer | In situ studies of macro- and microstresses in rocks | [10] |
| 5. | SKAT | The high-resolution texture diffractometer | Studies of texture of geological samples (rocks, minerals) | [11] |
| 6. | FSD | High resolution Fourier Stress Diffractometer | Determination of residual stresses in bulk industrial components and new advanced materials | [12] |
| 7. | DN-12 | Neutron Diffractometer for Investigations of Micro-Samples at High Pressure | Determination of parameters of crystal and magnetic structure of materials as function of external pressures | [13] |
| 8. | FSS | Fourier stress spectrometer (Diffractometer) | | |
| 9. | YuMO | Small-Angle Scattering | Determination of structural characteristics (size and shape of particles, agglomerates, pores, fractals) of nanostructured materials and nanosystems, including polymers, lipid membranes, proteins, solvents, etc. | [14] |
| 10. | REMUR | Polarized-neutron spectrometer | Determination of magnetization profile of layered magnetic nanostructures, studies of proximity effects in nanosystems | [15] |
| 11. | REFLEX | Polarized-neutron reflectometer | Determination of structural characteristics of thin films and layered nanostructures | [9] |
| 12. | GRAINS | Multifunctional neutron reflectometer with horizontal sample plane | Studies of surface and interface phenomena in soft and liquid nanosystems (magnetic fluids, polymers, lipid membranes) | [9] |
| 13. | DIN-2PI | Inelastic scattering | A study of lattice dynamics of crystalline, amorphous materials and liquids | [16] |
| 14. | NERA | Inelastic scattering | A study of lattice dynamics and structural parameters of molecular crystals, crystals with molecular ions, especially exhibiting polymorphism | [17] |
| 15. | NRT | Neutron radiography and tomography | Material research and studies of products for nuclear technologies, paleontological and geophysical objects, and objects of cultural heritage | [18] |
| 16. | REGATA | Radioanalytical complex | Determination of elemental composition of biological and geological objects | [19] |
| 17. | KOLKHIDA | Nuclear physics facility | Investigation of nuclear neutron precession and magnetic structure. | [9] |
| 18. | IZOMER | Nuclear physics facility | Determination of yields and decay constants of delayed neutron groups in minor actinide fission | |

The IBR-2 complex of spectrometers allows one to conduct a wide range of interdisciplinary research in the fields of condensed matter physics, materials science, chemistry, biology, geophysics, pharmacology, medicine, nuclear physics, ecology, etc.

Recently, a new iron oxide, $Fe_4O_5$, which can presumably exist in the layers of the Earth's upper mantle, has been synthesized under the combined effect of high pressures and temperatures.

A comprehensive study of its physical properties, as well as its atomic and magnetic structure using neutron diffraction techniques at the IBR-2 reactor, has revealed a new type of charge-ordering state with the formation of dimeric and trimeric electronic states in this compound (Figure 4) [20].

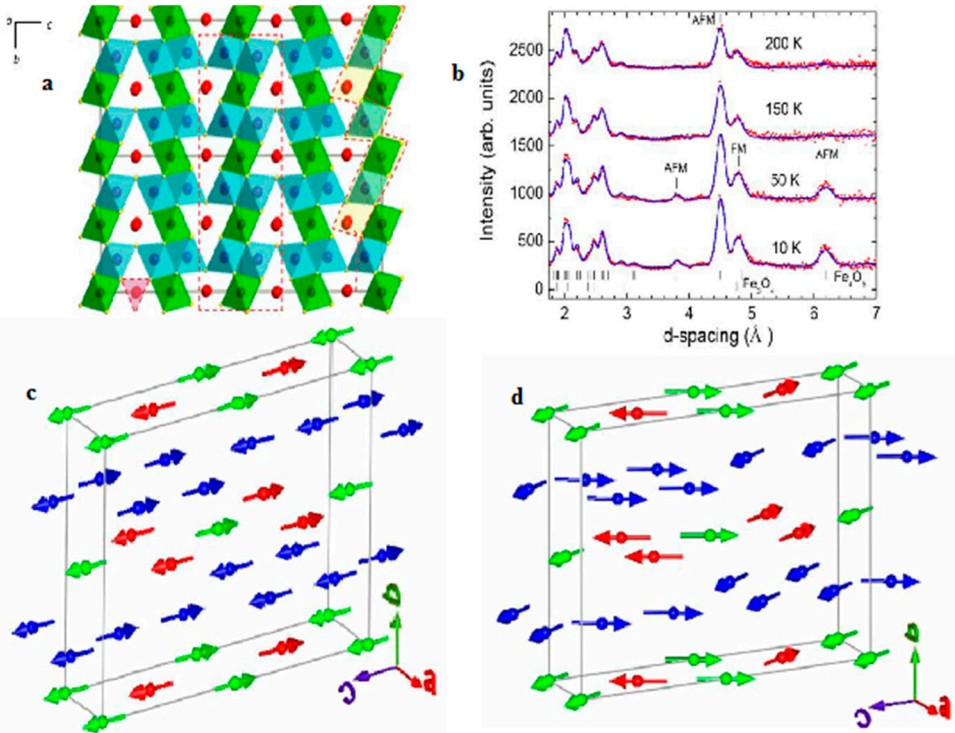

**Figure 4.** Crystal structure of $Fe_4O_5$ (**a**), neutron diffraction spectra obtained at different temperatures and treated by the Rietveld method (**b**), the magnetic structure at T = 150 K (**c**) and T = 10 K (**d**).

To improve the performance of lithium batteries, a series of experiments on neutron reflectometry (GRAINS reflectometer) to study the electrochemical interfaces of liquid electrolyte/solid electrode have been carried out. From the specular reflectivity analysis, the formation of a solid-electrolyte interphase (SEI) on the surface of the electrode (Cu) has been concluded, and the lithium electrodeposition and growth of parasitic dendritic structures during the operation of an electrochemical cell have been tracked [21].

## 3. IREN Facility

The IREN facility is an intense resonance neutron source developed for fundamental and applied investigations in neutron nuclear physics by precision neutron spectroscopy methods in a neutron energy range from eV to hundreds of keV. The IREN is based on an electron linear accelerator (LUE-200) with an S-band traveling wave accelerating structure. A massive target made of a material with a high atomic number (tungsten or uranium) serves as a source of neutrons. The electron beam produced by the LUE-200 linear electron accelerator hits the target and undergoes a conversion into neutrons (e-γ-n reaction). A conceptual diagram of the facility is shown in Figure 5.

The LINAC proposed by the Budker Institute of Nuclear Physics, Siberian Branch, Russian Academy of Sciences, consists of a pulsed electron gun, an accelerating system, a radio frequency (RF) power supply system based on 10-cm-range klystrons with modulators, and a beam focusing and transport system including a wide aperture magnetic spectrometer and a vacuum system. The accelerator is positioned vertically inside a three-story building. The LUE-200 electron source is a 2electrode electron gun with Ø 12-mm oxide thermocathode. The cathode is fed by a 200 kV pulsed transformer. The anode is the wall of the vacuum chamber with a hole 43 mm in diameter

closed with a wire frame made of stainless steel. The electron gun provides a pulsed electron beam with an 8 A peak current, 250–300 ns duration, 50 Hz repetition rate and ≤0.01π cm·mrad emittance. The accelerating system consists of an RF buncher and two short (3 m long) accelerating sections with a high acceleration rate. The accelerating section represents a circular blinded waveguide with constant impedance fed by the RF power. The RF power comes from klystrons—RF power amplifiers of a 10 cm frequency range (2856 MHz). The electron beam is focused by three short solenoidal magnetic lenses installed between the electron gun and the RF buncher. To measure the spectra of accelerated electrons, a broadband magnetostatic analyzer with a magnetic field perpendicular to the beam direction is used. A magnetic E-core with a variable magnetic rigidity of 0.166–0.7 T·m is placed after the second section. The electron beam in the spectrometer magnetic field is turned up to 90° and extracted from the spectrometer vacuum chamber through a 50 μm stainless steel window [22]. The neutron-producing target is made of tungsten-based alloy and represents a cylinder 40 mm in diameter and 100 mm in height housed within an aluminum can 160 mm in diameter and 200 mm in height. Distilled water is circulated inside the can, providing target cooling and neutron moderation. The thickness of the water layer in a radial direction is 50 mm. Time-of-flight measurements can be performed simultaneously on seven neutron beamlines. Several measurement stations are installed at different (10, 30, 50, 60, 85, 120, 250 and 500 m) distances along the flight paths. By the end of 2008, the first stage of the IREN facility comprising one accelerating section had been completed. The design parameters and parameters of the first and second stages are given in Table 3. During 2009–2015, the facility operated for physical experiments in the trial operation mode along with the ongoing modernization of some units and components of the accelerator. At the end of 2015, the second accelerating section with a new klystron and modulator was installed. At present, the facility composed of two accelerating sections is being adjusted and tested. The klystron technical parameters do not allow obtaining high electron energies and working with frequencies higher than 50 Hz.

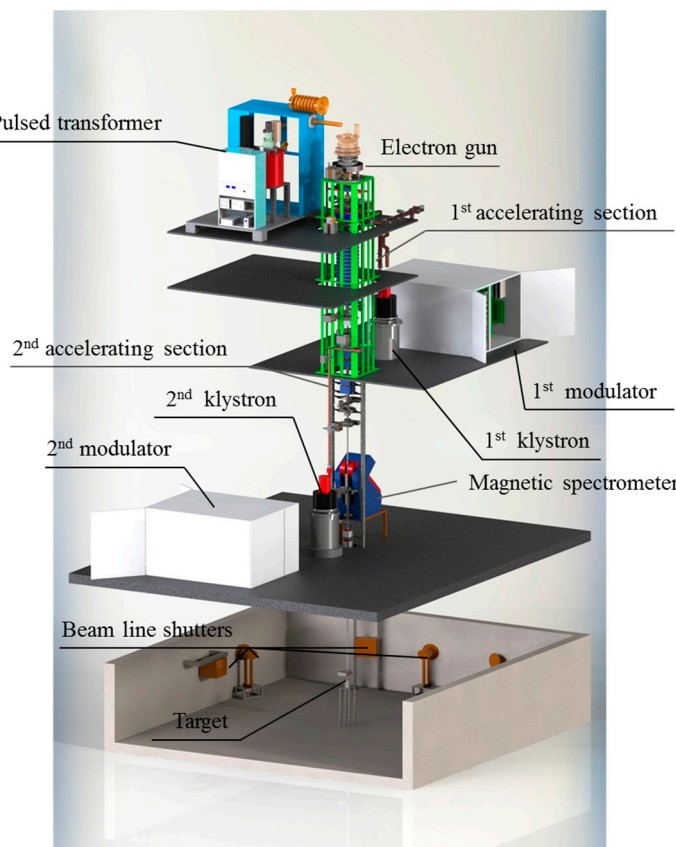

**Figure 5.** Layout of the IREN facility.

**Table 3.** IREN parameters.

| Parameter | Project | I Stage | II Stage |
|---|---|---|---|
| Peak current (A) | 1.5 | 1.5–2.5 | 1.5–2.5 |
| Repetition rate (Hz) | 150 | 25 | 50 |
| Electron pulse duration (ns) | 250 | 100 | 100 |
| Electron energy (MeV) | 212 | 32–42 | 45–65 |
| Beam power (kW) | 12 | 0.1–0.4 | 0.3–1.2 |
| Neutron intensity (n/s) | $2 \times 10^{13}$ | $3 \times 10^{11}$ | $6 \times 10^{11}$ |

The main research activities of the IREN facility include:

- Studies of T-odd, P-odd and P-even effects in the (n, $\gamma$), (n, p), (n, $\alpha$) reactions in the resonance neutron energy range;
- Studies of characteristics and correlations in the emission of neutrons, gamma quanta and light-charged particles in fission;
- Studies of total and partial neutron cross-sections, angular correlations, multiplicity fluctuations, yields of reaction products in neutron-nuclear interactions;
- Non-destructive analysis of elemental composition of objects using thermal, resonance and fast neutrons by neutron and gamma spectrometry;
- Development of experimental techniques.

In Table 4 the list of potential instruments in the IREN facility is presented.

**Table 4.** List of the IREN instruments.

| № | Name | Type | Main Feature | Domain of Research |
|---|---|---|---|---|
| 1 | ROMASHKA-1 | 12 NaI crystal system | Detection of $\gamma$-quanta | Determination of concentration of radioactive elements in the environment, study of radioactive neutron capture in the resonance neutron energy range |
| 2 | ROMASHKA-2 | 24 NaI crystal system | Detection of $\gamma$-quanta; movable, easy-adjustable | Study of radioactive neutron capture, $\gamma$-quanta multiplicity in resonances |
| 3 | | Large multi-sectional liquid scintillator detector | Detection of $\gamma$-quanta and neutrons; six sections, 210 l of scintillator | Nondestructive elemental/isotopic analysis by neutron resonance capture and neutron resonance transmission |
| 4 | | Ionization chambers | Different kinds: large target area; multi-sectional, solid and gaseous targets | Study of (n, p), (n, $\alpha$) reactions in the resonance neutron energy range |
| 5 | | Double section fission chamber with Frisch grid supplied with fast neutron scintillator detectors and $\gamma$-detectors | Measurements of the kinetic energy and masses of fission fragments, prompt fission neutrons, $\gamma$-rays from fission fragments | Study of fission in the resonance neutron energy range |
| 6 | REGATA-2 (in development) | Radioanalytical complex: irradiation channels, pneumatic transport setup, hpGe-detector, sample changer, equipment for sample preparation | Neutron and $\gamma$-activation analysis | Determination of elemental composition of biological and geological objects |
| 7 | | Neutron polarization setup (planned)) | | Study of the P-odd, P-even, T-odd effects in resonances |

**Conflicts of Interest:** The author declares no conflict of interest.

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
