# Peer review of "Neutron Sources at the Frank Laboratory of Neutron Physics of the Joint Institute for Nuclear Research"

_qubs, doi:10.3390/qubs1010006_

Reviewer 1 Report

The manuscript “Neutron Sources at Frank Laboratory of Neutron Physics of the Joint Institute for Nuclear Research ” by Shvetsov describes status and upgrades to the neutron sources at the Frank Laboratory. 

The two neutron producing facilities at this institute are described in some detail. The pulsed reactor IBR-2 has been upgraded and modernized. The most important changes and technical details are nicely presented.  Informative graphs illustrate the design of the pulsed reactor.  The second facility is based on an electron accelerator to produce high energy neutrons, which has just now been commissioned. This is a highly interesting and welcome review about opportunities for the neutron scattering community at the Frank Laboratory. The paper is well written beside a few points.

The manuscript is well adapted to the journal and I strongly recommend the publication of this manuscript in QBS after the following minor points are considered:

Cold moderator is mentioned for the IBR-2. Can you give some information about the performance of this new type of cold moderator.

Can you provide a few more details about the highlighting characteristics of the different instruments at the IBR-2.

Concerning table 2: Is it possible to add some details about the achieved performance after the second upgrade of the iren facility.

Some articles are missing:

Title: “.. at the Frank….”

Introduction: “The Joint Inst…..”

Author Response

The manuscript is well adapted to the journal and I strongly recommend the publication of this manuscript in QBS after the following minor points are considered:

Cold moderator is mentioned for the IBR-2. Can you give some information about the performance of this new type of cold moderator. Information was added, lines: 50-55

Can you provide a few more details about the highlighting characteristics of the different instruments at the IBR-2. The table with IBR-2 instruments was added (Table 2)

Concerning table 2: Is it possible to add some details about the achieved performance after the second upgrade of the iren facility. The column with IREN parameters achieved after second upgrade was added to Table 3

Some articles are missing:

Title: “.. at the Frank….” Done, line 2

Introduction: “The Joint Inst…..” We do not use article before JINR name

Reviewer 2 Report

This article gives an overview of the technical aspects and modes of operation of the two neutron sources at the Frank Laboratory of Neutron Physics (FLNP).  The pulsed reactor, IBR-2, is quite unique, and has been recently upgraded with small changes in the principal parameters, as listed in this paper.  The IREN intense resonance neutron source, which is in the early stages of construction (?), is possibly less well known than IBR-2, and it is very informative to have both facilities described in this paper.  What is lacking, in my opinion, is a little more information on the instruments on the two facilities and a couple of examples of recent scientific highlights.

I recommend publication in the Special Issue on Facilities of Quantum Beam Science if the following points are addressed:

Include a table of the 15 instruments (13 operating plus two under construction) on IBR-2 with, e.g. the name, instrument type, main feature, principal domain of research, and reference (if one exists).

Include a table of the 7 (planned?) instruments on IREN with, e.g. the name, instrument type, main feature, principal domain of research, and reference (if one exists).

Add brief descriptions of a couple of recent scientific highlights with a figure or two for each.

Lighten the lower part of Figure 4, and possibly use contrasting colour for the text in that part.

Comment on the low electron energy and beam power achieved thus far on IREN (Table 2.).

Please check the volume number for reference 6: This is given as 45 on the PEPAN website, although I could not find the article.

The paper is very well written, but I recommend a few minor changes, as follows: 

Line 9: Replace ‘are two main’ by ‘are the two main’.

Line 11: Replace ‘of IREN’ by ‘of the IREN’

Line 32: Replace ‘hexahedron’ by ‘hexagon’ (A hexahedron is a six-faced polyhedron, e.g. a cube.  The core has eight faces.)

Line 94: Replace ‘within aluminium’ by ‘within an aluminium’.

Line 120: Replace ‘1991, 2, 14-18’by ‘1991, 2(2), 14-18’. (The issue number should be given for Neutron News because the page numbering does not continue across issues.)

Author Response

Include a table of the 15 instruments (13 operating plus two under construction) on IBR-2 with, e.g. the name, instrument type, main feature, principal domain of research, and reference (if one exists). The table with IBR-2 instruments was added (Table 2)

Include a table of the 7 (planned?) instruments on IREN with, e.g. the name, instrument type, main feature, principal domain of research, and reference (if one exists). The table with IREN potential facilities was added (Table 4)

Add brief descriptions of a couple of recent scientific highlights with a figure or two for each. Done, lines 73-86

Lighten the lower part of Figure 4, and possibly use contrasting colour for the text in that part. The figure was changed.

Comment on the low electron energy and beam power achieved thus far on IREN (Table 2.). The information was added, lines 129-130

Please check the volume number for reference 6: This is given as 45 on the PEPAN website, although I could not find the article. The article is available at https://link.springer.com/article/10.1134/S1547477114050069

The paper is very well written, but I recommend a few minor changes, as follows: 

Line 9: Replace ‘are two main’ by ‘are the two main’. Done, line 9

Line 11: Replace ‘of IREN’ by ‘of the IREN’ Done, line 11

Line 32: Replace ‘hexahedron’ by ‘hexagon’ (A hexahedron is a six-faced polyhedron, e.g. a cube.  The core has eight faces.) Done, line 33

Line 94: Replace ‘within aluminium’ by ‘within an aluminium’. Done, line 1119

Line 120: Replace ‘1991, 2, 14-18’by ‘1991, 2(2), 14-18’. (The issue number should be given for Neutron News because the page numbering does not continue across issues.) Done

Round  2

Reviewer 2 Report

I thank the author for considering my comments on the first version and especially for including the two tables listing the instruments on the IBR-2 reactor and the IREN facility.

There are just some minor inconsistencies between the Table 2 and Figure 3 that I suggest be clarified.

1) Figure 3: KOLKHIDA, REGATA, and IZOMER are not listed in Table 2. Perhaps simply note why in the caption.

 2) Table 2, No. 2: RTD is not indicated in Figure 3.  I presume that it is denoted DN-2 in the figure.  If so, change the table entry to ‘RTD (DN-2)’.

In addition I note the following typographical corrections:

3) Line 48: Replace ‘Joint Institute’ by ‘The Joint Institut’

 4) Line 51: Replace ‘Two main JINR neutron sources – fast’ by ‘The two main JINR neutron sources – the fast’

 5) Line 52: Replace ‘and intense’ and ‘and the intense’

 6) Line 57: Replace ‘shutdown’ by ‘shut down’

 7) Line 110: Replace ‘Other two’ by ‘Two other’

 8) Table 2, No. 5: Replace ‘minerals’ by ‘minerals)’

 9) Table 2, No 10: Replace ‘Spectrometer of polarized neutrons’ by ‘Polarized-neutron spectrometer’

 10) Table 2, No 11: Replace ‘Reflectometer of polarized neutrons’ by ‘Polarized-neutron reflectometer’

 11) Line 145: The formula Fe4O5 lacks subscripts.

 12) Table 4, No 6: Replace ‘Radioanalitical’ by ‘Radioanalytical’

 13) Lines 252, 258, and 265: The journal names should be in italics.

Author Response

There are just some minor inconsistencies between the Table 2 and Figure 3 that I suggest be clarified.

 1) Figure 3: KOLKHIDA, REGATA, and IZOMER are not listed in Table 2. Perhaps simply note why in the caption. [ KOLKHIDA, REGATA, and IZOMER were added in the Table 2]

 2) Table 2, No. 2: RTD is not indicated in Figure 3.  I presume that it is denoted DN-2 in the figure.  If so, change the table entry to ‘RTD (DN-2)’. [Changed in the Table]

In addition I note the following typographical corrections:

 3) Line 48: Replace ‘Joint Institute’ by ‘The Joint Institut’ [Changed, line 17]

 4) Line 51: Replace ‘Two main JINR neutron sources – fast’ by ‘The two main JINR neutron sources – the fast’ [Changed , line 20]

 5) Line 52: Replace ‘and intense’ and ‘and the intense’ [Changed, line 21]

 6) Line 57: Replace ‘shutdown’ by ‘shut down’ [Changed, line 26]

 7) Line 110: Replace ‘Other two’ by ‘Two other’ [Changed, line 63]

 8) Table 2, No. 5: Replace ‘minerals’ by ‘minerals)’ [bracket was added]

 9) Table 2, No 10: Replace ‘Spectrometer of polarized neutrons’ by ‘Polarized-neutron spectrometer’  [Changed]

 10) Table 2, No 11: Replace ‘Reflectometer of polarized neutrons’ by ‘Polarized-neutron reflectometer’

 [Changed]

11) Line 145: The formula Fe4O5 lacks subscripts. [Done] 

12) Table 4, No 6: Replace ‘Radioanalitical’ by ‘Radioanalytical’ [Changed]

 13) Lines 252, 258, and 265: The journal names should be in italics. [Done]
